# Potential Early Adopters of Hybrid and Electric Vehicles in Spain—Towards a Customer Profile

**Elena Higueras-Castillo [1], Sebastian Molinillo [2]⬤, J. Andres Coca-Stefaniak [3],*⬤ and Francisco Liébana-Cabanillas [1]⬤**

1   Department of Marketing and Market Research, Faculty of Business and Economics, University of Granada, Paseo de Cartuja 7, 18011 Granada, Spain; ehigueras@ugr.es (E.H.-C.); franlieb@ugr.es (F.L.-C.)
2   Department of Business Management, Faculty of Economics and Business, University of Malaga, El Ejido Campus, 29071 Malaga, Spain; smolinillo@uma.es
3   Department of Marketing, Events and Tourism, Faculty of Business, University of Greenwich, Park Row, London SE10 9LS, UK
*   Correspondence: a.coca-stefaniak@gre.ac.uk; Tel.: +44-208-331-8309

**Abstract:** The adoption of electric vehicles (EVs) by consumers is regarded as a key strategic goal for the reduction in transport-related air pollution levels. Although sales of EVs continue to rise year on year, generally, the attainment of the strategic goals set by various governments for the adoption of EVs remains a distant target. The purpose of this study is to identify the customer profile of early adopters of EVs in Spain: one of Europe's largest economies, yet the country with the lowest uptake of EVs at present. The analysis presented here is based on an online survey of 404 potential consumers of EVs in Spain. A cluster analysis of the data was performed based on a set of three socio-demographic characteristics (gender, age, and income), one psychographic (green moral obligation—GMO) and two EV attributes (price and driving range). The results of this analysis showed that there exist two segments with respect to higher or lower customer intentions related to the adoption of EVs. These findings represent a theoretical contribution to current understanding of the customer profile of adopters of EVs and will contribute to the development of communication and retail strategies aimed at customers fitting the profile of early adopters of new technologies.

**Keywords:** electric vehicles; early adopters; cluster analysis; segmentation; sustainable mobility

## 1. Introduction

In 2019, the UN Climate Change Conference held in Madrid (Spain) called for a prioritisation of energy policies that support a transition towards low-carbon solutions in view of the continued rise in $CO_2$ emissions, which have been linked to global warming [1]. In Europe, transport is responsible for approximately one quarter of greenhouse gas emissions and remains the main source of air pollution in cities. Furthermore, road transport accounted for 70% of these emissions in 2014 [2]. However, the type of transition required towards more sustainable models of energy consumption implies considerable changes for this sector of the economy. In line with this, electric vehicles (EVs) remain at the forefront of new transport technologies with regards to reducing pollution and greenhouse gas emissions [3]. In this study, EV refers to battery-operated electric cars (i.e., solely electric vehicles) and plug-in hybrid electric cars (i.e., those that combine an internal combustion engine system with electric propulsion).

The EV market is growing rapidly. From 2013 to 2018, the global fleet of electric passenger vehicles rose tenfold to surpass five million vehicles in 2018, of which 45% were in China, 24% in Europe and 22% in the United States [4]. However, despite this level of growth, the market share of EVs remains modest compared to that of internal combustion engine vehicles. For instance, EV sales as a fraction of all new car sales in Europe represented only 6.3% of the market in the third quarter of 2018 [5], even if there are considerable country differences across Europe in terms of market share [6] as well as key factors affecting consumer behaviour [7]. Therefore, a country-level analysis is necessary to elicit the factors having the largest impact on the adoption of EVs in each case.

To date, the majority of published research has focused on countries like China, the United States, Germany or the Scandinavian region, whilst only a limited number of studies exist on key economies where the market share of EVs is low, such as Spain—the fifth largest country in Europe in terms of population, and an important global supplier of vehicles. In Spain, the transport sector generated 26.1% of all noxious gas emissions in 2017 [8]. In line with this, electromobility would not only appear to be a desirable option in Spain from a public health and environmental perspective but could also be linked to a source of competitiveness for this country in world markets with positive impacts on the country's economy and levels of employment. Yet, although the Spanish government has prioritised an increase in the national EV fleet by 2030 as part of its policies [8], Spain's current share of the market for EVs remains one of the lowest in Europe [6]. Even when annual sales of EVs nearly doubled from 2017 to 2018, this trend would not suffice to achieve the policy goals set out in this respect in the National Integrated Strategy for Energy and Climate [8], with Spain lagging on this front well behind the European average [9]. The global electromobility index in the third quarter of 2019 positioned Spain in the penultimate place European Union countries, with one of the lowest rates of market growth in 2019 [10]. Therefore, a better understanding of the characteristics of potential adopters would support better decision making among policy makers and business managers, especially with respect to relevant marketing initiatives for EVs.

Research grounded in innovation theory [11] posits that there are different typologies of potential adopters of EVs, which are often linked to the time lapse between the product's launch and its adoption. Today, EVs remain in the early stages of the adoption in Spain, which raises the importance early adopters in this process as this typology of consumers tends to adopt new technologies faster than market segments [12–14]. As a result, this market segment remains key to research on EVs and has been analysed in various countries around the world. In order to define the characteristics of early adopters, scholars have used a wide array of factors that can be clustered into two key categories, namely individual-related variables and EV attributes [15]. Common examples of the first category include age, gender, income, education level [16,17], social influence, environmental concern, and innovativeness [18,19]. Attributes used in earlier research studies to analyse consumer behaviour related to the adoption of EVs include price [20], driving range [21], availability of charging points [22] and vehicle acceleration [23].

The purpose of this study is to establish a profile of early adopters of EVs in Spain. Further, the study's focus is on the identification and description of market segments in this respect using clustering algorithms. On this front, Loker and Perdue [24] suggest the use of a combination of descriptive variables (e.g., demographic) along with predictive factors (e.g., advantages or benefits obtained by customers from specific products), as users in identical demographic groups can display varying behaviours depending on their underlying motivations. Following on from a review of the literature and in line with rational choice theory, this study posits that the adoption of EVs by consumers revolves around two specific attributes—price and driving range. In turn, this choice is influenced by four individual-related variables, three of which involve socio-demographic characteristics (i.e., gender, age and income), with the fourth one—green moral obligation (GMO)—based on a psychographic approach. Research has found that these four variables are the most influential in the adoption of EVs [15,25,26]. In order to evaluate the effect of these factors, this study gathered consumer data using

an online survey, which rendered 404 responses of potential adopters of EVs. The resulting data was then subject to a cluster analysis.

In line with this, the novelty of this study rests not only on the lack of published research related to the characteristics of potential early adopters of EVs in key economies where EV's still have a low market share, but also in the fact that this is the first study of its kind in Spain in seeking to establish the profile of early adopters of these vehicles at an early stage in their market entry. Similarly, this is the first research study to evaluate the effect of GMO on the adoption of EVs. Although earlier studies have included pro-environmental behaviour as a variable, the concept of GMO taps into more deeply rooted beliefs and personal values. Moreover, few earlier studies have considered the joint effect of different variables taking into consideration consumers' socio-demographic and psychographic characteristics along with technical and financial factors affecting consumer choice.

Next, a revision of the literature is carried out setting out the theoretical context of this study. Then, the research methodology is explained, including the method used in the analysis of the data gathered. Following on from that, the results of the research are outlined and the implications of the findings discussed in the context of theory and practice with regards to the adoption of EVs. Finally, this article sets out the limitations of this study along with future areas of further research.

## 2. Theoretical Background

### 2.1. The Role of Individual-Related Characteristics in the Adoption of EVs

The literature on the adoption of EVs to date has identified a wide range of factors influencing consumer decision making across different countries. For instance, Peters and Dütschke [27] found that early adopters of EVs in Germany were more likely to be middle-aged men living in a household with ownership of several other vehicles. Similarly, research by Hardman et al. [12] in the USA showed that consumers most likely to purchase EVs tended to have already 2.5 other vehicles, which is above that country's average in terms of individual car ownership. On the other hand, research by Plötz et al. [14] in Germany found that early adopters of EVs in that country tended to live in rural areas, whereas urban dwellers were less likely to adopt EVs on the grounds of limited driving range. In China, Zhang et al. [28] found that educational attainment, number of dependants and the number of vehicles owned in each household were key factors affecting the perceived affordability of EVs among different consumer segments.

Now, of all potential variables affecting consumer behaviour, scholarly research on the adoption of EVs shows that adopters' socio-demographic characteristics can be important predictors of adoption behaviour. Although there is no consensus among scholars on the profile of adopters of EVs, some authors have argued that consumer behaviour in this respect depends on socio-demographic characteristics such as gender, age and income (see Table 1). In fact, most studies considering gender as a variable have found that men are more prone to adopting EVs at the product's earlier market entry stages [12,14,29,30]. This may be due in part to a stronger marketing emphasis at these stages on issues such as mechanics, engineering or driving experience, which often tend to have higher levels of appeal to men. However, other studies have found that women are more prone to be early adopters of EVs [31–34] due to their generally higher level of environmental awareness. On the other hand, other research studies have argued that when other characteristics are considered, both men and women are similarly prone to adopting EVs [21]. Therefore, although there is no consensus among scholars on a specific gender bias with regards to the adoption of EVs, it is widely recognised that gender issues play a role in the overall decision-making process.

Similarly, consumers' age is another characteristic often considered when analysing the marketing of EVs. Most research to date concurs that younger people or people below the age of 45 are more prone to purchasing EVs [18,26,32,34]. This may be because younger people are more prone to trying new products, have a more favourable attitude towards change as well as a higher level of environmental consciousness [35,36]. However, a more modest number of research studies have found that older

people tend to be more predisposed to adopt EVs [37,38], possibly due to being less price-sensitive in their purchases and potentially less concerned about restrictions related to these vehicles' driving range compared to existing combustion engine alternatives [39]. Regardless, age was found to be a significant factor in the literature with regards to customers' decision to adopt EVs.

Traditionally, innovation research has shown that early adopters tend to enjoy a higher socio-economic standing than late adopters. This is often as a result of their generally higher level of education and income as well as product knowledge [11]. In the context of EVs, existing research shows that income and social status tend to influence positively the product's adoption [40–43], even if a handful of studies have questioned this assertion as wealthier individuals may in some cases favour luxury combustion engine vehicles as a status symbol over EVs [33]. All in all, income is a key variable influencing the adoption of EVs.

**Table 1.** Socio-demographic characteristics of early adopters of EVs.

| Source | Country | Gender | | Age | | Income/Social Status | |
|---|---|---|---|---|---|---|---|
| | | Female | Male | ≤45 Years of Age | >45 Years of Age | Lower than Late Adopters | Higher than Late Adopters |
| Curtin et al. [41] | USA | | | | | | √ |
| Erdem et al. [30] | Turkey | | √ | √ | | | √ |
| Hidrue et al. [33] | USA | | | √ | | | |
| Ozaki and Sevastyanova [44] | UK | | | √ | | | |
| Campbell et al. [40] | UK | | | | | | √ |
| Deloitte [45] | USA | | | √ | | | √ |
| Egbue & Long [29] | USA | | √ | | | | |
| Hackbarth and Madlener [18] | Germany | | | √ | | | |
| Peters and Dütschke [27] | Germany | | √ | | | | |
| Plötz et al. [14] | Germany | | √ | | | | |
| Ziefle et al. [34] | Germany | √ | | | | | |
| Kawgan-Kagan [31] | Germany | √ | | | | | |
| Trommer et al. [43] | Germany | | √ | √ | | | √ |
| Axsen et al. [46] | Canada | | | | | | √ |
| Hardman et al. [12] | USA | | √ | | | | √ |
| Mohamed et al. [16] | Canada | | | √ | | | |
| Javid and Nejat [47] | USA | | | | | | √ |
| Wang et al. [33] | China | √ | | √ | | √ | |
| She et al. [38] | China | | | | √ | | |
| Vassileva and Campillo [17] | Sweden | | √ | | | | √ |
| Lin & Wu [32] | China | √ | | √ | | | |
| Rodríguez-Brito et al. [42] | Spain | | | | | | √ |
| Sovacool et al. [21] | Nordic countries | √ | √ | √ | | | √ |
| Araújo et al. [48] | USA | | | | | | √ |
| Chen et al. [49] | Nordic countries | | √ | √ | | | √ |
| Lee et al. [13] | USA | | | | | | √ |
| Oliveira and Dias [37] | Portugal | | | | √ | | |
| Sovacool et al. [50] | Nordic countries | | √ | | | | |

Source: own conception.

Lastly, along with the three socio-demographic characteristics outlined earlier, this study also considers psychographic characteristics as influential on an evaluation of the profile of early adopters of EVs. Generally, customer values tend to have a key effect on the adoption of sustainable technologies, particularly when customer behaviour is led by normative goals. In other words, when the adoption of a product is governed by individual preferences related to the product impact on society and/or the environment [51]. Research in this context has shown that the adoption of EVs can be swayed by consumers' environmental concerns, so the decision-making process will be influenced by personal norms related to environmentally friendly behaviours [15]. In line with this, this study explores the influence of GMO on consumers' decision making. The definition of GMO adopted here is "the extent to which an individual feels a sense of responsibility to act (or not) morally (or immorally) when faced with an ethical situation" [52]. Earlier research has shown that when consumers see themselves as "green" or "eco-friendly", there is a higher likelihood that they will feel a moral obligation towards making more ethical and environmentally conscious decisions [53–57]. For instance, Wang et al. [33] showed that personal moral norms have a moderating effect on consumers' environmental concerns regarding their intention to adopt EVs. Moreover, the literature on this topic shows that potential adopters of EVs tend to be more concerned about environmental issues [18,26,38] and believe that EVs generally have a lower negative impact on the environment overall than other types of vehicles [19,32,45]. One of the most influential attributes on the adoption of EVs is their lower $CO_2$ emission levels compared to internal combustion engine vehicles [17,18,50,58] as well as other general environmental attributes [59]. Therefore, this study interprets GMO as a key characteristic among consumers, which may contribute to a better definition of the market profile of early adopters of EVs.

## 2.2. The Role of Financial and Technical Attributes in the Adoption of EVs

In a similar manner to individual-related characteristics, earlier research incorporates a wide range of perspectives in its analysis of the financial factors and technical characteristics affecting the adoption of EVs by consumers. Whereas some scholars have noted that EVs have certain technical advantages such as recharge and maintenance costs, which are key factors in their adoption [44], research by Zhang et al. [60] showed financial benefits as the key factor determining willingness to purchase of EVs by consumers. Yet, in spite of cost savings, consumers may still be reluctant to purchase EVs in line with the energy efficiency paradox or energy efficiency gap posited by some scholars [61]. In line with this, some studies have shown that some consumers either do not place a great deal of value on these benefits or are not aware of the potential cost savings involved [62]. In fact, some consumers are more influenced by high purchase prices and do not take into account the product's life costs [63]. Given this situation, a number of governments have recently implemented different policy interventions to raise awareness of electric vehicles [64]. Scholarly studies have shown that these incentives vary across countries, including the USA [65], Europe [66,67] and elsewhere [68]. Among other policy interventions, Egnér and Trosvik [69] point out that local incentives such as infrastructure development can also have a significant impact on EV adoption rates. In this respect, recharge point infrastructure is an essential factor in this process as its lack of development can be a major obstacle in the adoption of EVs [70].

Research published to date shows that vehicle price and driving range are two of the most dominant attributes affecting the adoption of EVs [15,26]. In this respect, price is defined as the monetary value attached to the product by its seller and the resulting money disbursed on that product by the customer. Although the price of EVs is generally higher than that of most comparable alternatives [59], some research studies have shown that concern for the environment influences some consumers' willingness to pay these higher prices [71], though other studies have contradicted these findings [72]. In addition to this, several studies have found that price is a barrier to many potential adopters [25,73,74]. Similarly, perceived value for money has also been found to have a significant positive effect on consumers' attitude towards EVs [20], whilst earlier research has shown that financial incentives [31] or a significant reduction in market prices would encourage the adoption of these

vehicles by potential consumers [75–77]. Moreover, Tran et al. [78] found that the most influential factor among early adopters of EVs is the financial benefit. Therefore, price perception may be deemed a significant factor to consider when eliciting a profile for early adopters of EVs.

Parallel to this, driving range represents one of the key barriers to the adoption of EVs [15]. Generally, consumers perceive the overall autonomy of EVs currently as very limited [79,80], with EVs' driving range deemed as a negative factor by the majority of consumers evaluating this product [19], with a significant proportion of consumers displaying anxiety with regards to the risk of the vehicle's battery getting depleted during journeys [38]. Therefore, an increase in the driving range of EVs would appear to have a positive influence on their adoption by consumers [23], even if this factor's actual influence may depend on the needs of specific consumers, given that those who either need them to cover shorter distances and/or have another vehicle for longer trips may be less influenced by this issue [22,81].

Finally, it is worth noting that research on the adoption of EVs in Spain remains embryonic. For instance, Junquera et al. [82] used logistic regression analysis to show the influence of various variables (e.g., price, range, age) on customers' willingness to purchase an electric car. Subsequently, Martínez-Lao et al. [83] analysed and assessed the charging systems of EVs in Spain, arguing the need for the development of relevant policies to extend the network of charging points. More recently, Rodríguez-Brito et al. [42] identified certain characteristics of early adopters of EVs (e.g., range, environmental awareness, risk behaviour) among consumers on the Spanish island of Tenerife. However, to the best of the authors' knowledge, no published scholarly study has analysed the profile of early adopters in Spain using a representative sample of the whole country. As a result, this study provides a novel contribution to existing knowledge on the adoption of EVs by improving our understanding of the characteristics of potential adopters at an early stage in the introduction of EVs in the Spanish market.

## 3. Research Method

### 3.1. Data Collection

Data for this study was collected using an online survey from April to July of 2018. For this purpose, a consumer database provided by a market research company, Toluna, was used. This market research company manages a panel of consumers representing countries such as the USA, Germany, Japan and Spain, among others. The authors of this study engaged the services of this market research company to administer the survey to a sample of Spanish consumers over the age of 18 and in possession of a valid driving license. The company collected the data using an ad hoc questionnaire provided by the authors of this study, so that the resulting database obtained is private and original in terms of its content. Once invalid responses were eliminated, a sample of 404 valid responses was obtained. A sampling error of 4.874% was achieved with a confidence interval of 95%. Table 2 outlines the characteristics of this study's sample. Fifty-one percent of respondents were male, with 60.2% of respondents below the age of 46 and 44.1% educated to university degree level; 58.7% were in full-time employment and 56.9% had a monthly income in the range of 1100 to 2700 EUR; 76.9% of respondents had more than five years of driving experience, with 61.6% driving over 12,500 km per year. Therefore, the sample fits the overall statistical distribution of the population of Spain well in terms of gender, even if younger market segments are over-represented as opposed to older segments beyond the age of 65, which only account for 7.7% of the sample, when the proportion of people over that age represents 22% of Spain's population. This difference in the sample's age distribution was influenced by the study's focus on consumers more likely to adopt EVs, in line with earlier similar studies (see Table 1).

**Table 2.** Demographic characteristics of respondents (*n* = 404).

| Variable | Description | Frequency | % of Sample |
|---|---|---|---|
| Gender | Female | 206 | 51.0 |
| | Male | 198 | 49.0 |
| Age | 18–25 | 55 | 13.6 |
| | 26–35 | 111 | 27.5 |
| | 36–45 | 77 | 19.1 |
| | 46–55 | 74 | 18.3 |
| | 56–65 | 56 | 13.9 |
| | Older than 65 | 31 | 7.7 |
| Education | Basic schooling or less | 21 | 5.2 |
| | Vocational training | 114 | 28.2 |
| | University bachelor's degree | 178 | 44.1 |
| | Postgraduate degree | 91 | 22.5 |
| Employment status | Unemployed | 37 | 9.2 |
| | Student | 43 | 10.6 |
| | Employed | 237 | 58.7 |
| | Self-employed | 39 | 9.7 |
| | Retired | 48 | 11.9 |
| Monthly income (Euros) | No income | 37 | 9.2 |
| | Less than 1100 EUR | 73 | 18.1 |
| | From 1100 to 1800 EUR | 135 | 33.4 |
| | From 1800 to 2700 EUR | 95 | 23.5 |
| | More than 2700 EUR | 40 | 9.9 |
| | Don't know/No answer | 24 | 5.9 |
| Driving experience (years) | 0–1 | 27 | 6.7 |
| | 1–3 | 34 | 8.4 |
| | 3–5 | 32 | 7.9 |
| | 5–8 | 207 | 51.2 |
| | More than 8 | 104 | 25.7 |
| Annual distance driven (km) | Up to 2500 | 78 | 19.3 |
| | Up to 7500 | 77 | 19.1 |
| | Up to 12,500 | 75 | 18.6 |
| | Up to 15,000 | 57 | 14.1 |
| | Up to 20,000 | 64 | 15.8 |
| | Up to 32,500 | 31 | 7.7 |
| | More than 32,000 | 21 | 5.2 |

Source: own conception.

### 3.2. Measurement

The survey questionnaire contained a brief outline of the objectives of the research as well as the contact details of the study's lead researcher, with full anonymity guaranteed to all respondents. All questions asked were closed in format and it was made clear to respondents that all answers were subjective in nature with no right or wrong answer options. The survey questionnaire was divided into three distinct sections: in the first section, a filter question was included with a number of other control questions, including "do you have a valid driving license for cars?", "do you own a vehicle?", "are you aware of the existence of electric vehicles?", "do you own an electric vehicle?", "have you ever considered purchasing an electric vehicle?". Following on from these questions, participants in the survey were asked about potential factors influencing their intention to purchase an electric vehicle. Finally, the third section of the questionnaire included questions related to respondents' socio-demographic data. This research was conducted as part of a much larger study, so the variables outlined below only correspond to those used for the segmentation of potential adopters of EVs.

The scales used for the measurement for variables such as GMO, driving range, price and intention were adopted from earlier studies (Table 3). GMO was measured using a scale used by Barbarosa et al. [53] and Sparks and Shepherd [84], whilst the scale for driving range was based on work by Schmalfuß et al. [85]. In order to evaluate the influence of price as a variable, a scale used by He and Zhan [86] and Petrick [87] was implemented in this study. Finally, the intention to adopt EVs by consumers was measured using three items used previously in research by Barbarrosa et al. [53] and Moons and Pelsmacker [88]. All four variables were measured using a 7-point Likert scale where 1 represented "fully disagree" and 7 stood for "fully agree".

**Table 3.** Measurement scales.

| Variable | Items | Source |
|---|---|---|
| GMO | I would feel guilty if I drove a car that knowingly damaged the environment. | Barbarossa et al. [53]; Sparks and Sheptherd [84] |
| | To buy a car that damages the environment would be morally wrong for me. | |
| | Buying a car that has a negative effect on the environment would go against my principles. | |
| Driving range | I feel uncomfortable with the limited driving range of EVs. | Schmalfuß et al. [85] |
| | The average driving range of EVs is not satisfactory. | |
| | The driving range of EVs is not suitable for my daily mobility requirements. | |
| | Due to the limited driving range of EVs, I would feel that my freedom to roam is restricted. | |
| Price | EVs are expensive. | He and Zhan [86]; Petrick [87] |
| | EVs are unaffordable. | |
| | The overall price of EVs is higher than that of similar combustion engine vehicle alternatives. | |
| | The price of EVs is higher than I expected. | |
| Intention to adopt | Next time I buy a car, I will consider buying an EV vehicle. | Barbarrosa et al. [53]; Moons and De Pelsmacker [88] |
| | In the near future, I expect to drive an EV car. | |
| | I intend on driving an EV vehicle in the near future. | |

Source: own conception.

### 3.3. Data Analysis

As a preliminary step before initiating the cluster analysis, items related to psychographic and EV attributes was analysed. Firstly, the reliability of the constructs was assessed using Cronbach's alpha [89]. Table 4 shows that Cronbach's alpha was greater than 0.7 for all of the constructs. Therefore, the internal consistency of the measures was good. Secondly, a factor analysis was performed in order to rule out the problem of multiple measures of similar constructs by identifying the latent variables. In line with this, a principal component analysis was carried out. The analysis was subjected to VARIMAX rotation, which reduced moderate factor loadings and increased higher factor loadings, resulting in clear factor loadings (see Table 4). From this analysis, three factors were identified on the basis of criterion eigenvalues higher than 1.

Then, in order to evaluate the role of the variables chosen for this study on the profile of early adopters of EVs, a cluster analysis was performed. Cluster analysis is a technique that enables heterogeneous data to be classified into subgroups of cohesive objects and evaluate the level of separation between them. This allows to quantify the degree of similarity, where a higher degree of proximity exists, as well as differences, where the distances between objects are larger. This renders homogeneous (cohesive) classification sets [90].

<div align="center">**Table 4.** Reliability.</div>

| Constructs | Items | Mean | SD | Factor Loadings | Cronbach's Alpha |
|---|---|---|---|---|---|
| GMO | GMO1 | 4.63 | 1.685 | 0.904 | 0.933 |
| | GMO2 | 4.57 | 1.702 | 0.940 | |
| | GMO3 | 4.49 | 1.728 | 0.918 | |
| Driving range | DR1 | 4.65 | 1.755 | 0.817 | 0.781 |
| | DR2 | 3.97 | 1.624 | 0.566 | |
| | DR3 | 3.70 | 1.722 | 0.657 | |
| | DR4 | 4.60 | 1.759 | 0.870 | |
| Price | PRI1 | 5.23 | 1.490 | 0.885 | 0.896 |
| | PRI2 | 5.25 | 1.452 | 0.914 | |
| | PRI3 | 5.47 | 1.443 | 0.841 | |
| | PRI4 | 5.03 | 1.475 | 0.812 | |

<div align="center">Source: own conception using SPSS software.</div>

The cluster analysis carried out in this study involved a two-phase conglomerate analysis. This allowed for the dataset to be explored initially by eliciting natural clusters emerging from the data itself. The process involved two stages, namely [91,92]:

1. Pre-cluster development. At this stage, a new data matrix was developed with a lower number of cases in preparation for the next stage. This was done by developing a tree of conglomerate characteristics (dendogram), with the resulting pre-clusters used as new cases.
2. Classification of pre-clusters. This stage involved the development of clusters from the earlier cases obtained. In order to do this, a hierarchical clustering approach was used.

The distances obtained between objects determined the manner in which the similarity between both conglomerates was calculated. This study used continuous and categoric variables, so the log-likelihood function was used for the analysis of the data as this function integrates both types of variables to establish probability-based distances between objects. The number of clusters was not pre-determined and instead it was allowed to emerge automatically using the Bayesian Information Criterion (BIC) [93]. The IBM SPSS Statistics 20 software was used for this analysis.

As part of the data analysis, a dependent variable (customers' intention to adopt EVs) and a set of independent variables were used. The independent variables included gender, age, income, GMO, vehicle driving range and price. Age was coded using two categories: young and old. Income was coded using three categories: low, medium and high. On the other hand, variables such as GMO, range and price are continuous and include 3, 4 and 4 items each, respectively. Therefore, a new variable was developed for each of them building on the mean values obtained for each item, i.e., mean GMO (4.5644), mean range (4.2327) and mean price (5.2438). As a result of this, the mean of each variable was used as the boundary to recode and classify the original variables into two groups: low and high (see Table 4). This method has been used and validated in earlier studies [94,95]. Figure 1 shows the recoding performed as a result of this process.

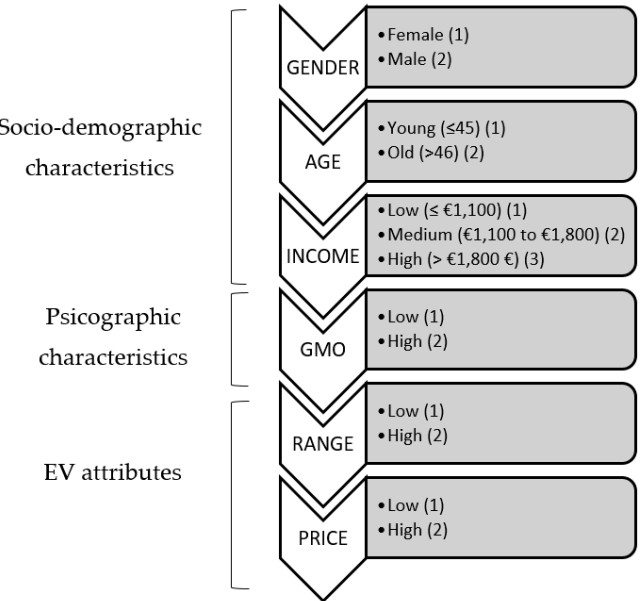

**Figure 1.** Segmentation of variables used in this study. Source: own conception.

## 4. Results

After applying the algorithm in two stages to the seven variables outlined above, two clusters were obtained with a cluster quality (cohesion and separation) of "fair" reported by the software [96]. The size of the clusters was very balanced with the first cluster formed by 189 respondents (49.7%) and the second one containing 191 respondents (50.3%). The size ratio between clusters was 1.01.

The importance of predictor variables is shown in Figure 2, with vehicle driving range emerging as the most important variable (importance of 100%) for the classification of respondents and the creation of segmentation profiles. This variable was followed by GMO and price, with levels of importance of 6% and 5%, respectively.

Based on the level of adoption, as the variable under scrutiny here, two groups were defined—one with the highest probability of adopting an EV and the other one with the lowest probability of this event occurring.

1.  The higher probability group accounted for 49.7% of the entire sample. The mean value of this group's intention to adopt an EV was 5.40/7.
2.  The lower probability group included the remaining 51.3% of the overall sample in this survey. In this case, the mean value of intention to adopt an EV was 4.52/7.

The first cluster can be characterised socio-demographically as formed by female respondents (53%), young respondents (52.4%) and individuals with high incomes (37.6%). In addition to this, the individuals from this group had a high GMO (65.1%) and had generally less negative perceptions related to driving range (97.9%) and price (58.2%) than in the case of the second cluster. The average intention to adopt EVs of individuals in this cluster was of 5.4 out of a maximum possible of 7. Therefore, it appears that this segment of the respondent sample would be more favourable towards the adoption of EVs.

In the second cluster, the prevailing profile includes mostly male respondents (51.3%), younger people (69.6%) and respondents with a medium income level (39.3%). Most respondents in this cluster had a low GMO (57.6%), and a very negative perception of EVs' limited driving range (100%) as well as their price (61.8%). The average intention to adopt EVs among respondents in this cluster was of 4.52 out of a maximum of 7. All in all, this cluster would appear to include primarily respondents with a low intention of adopting EVs. Table 5 summarizes the composition of each variable in each group and Figure 3 illustrates visually the characteristics of these two clusters using SPSS software.

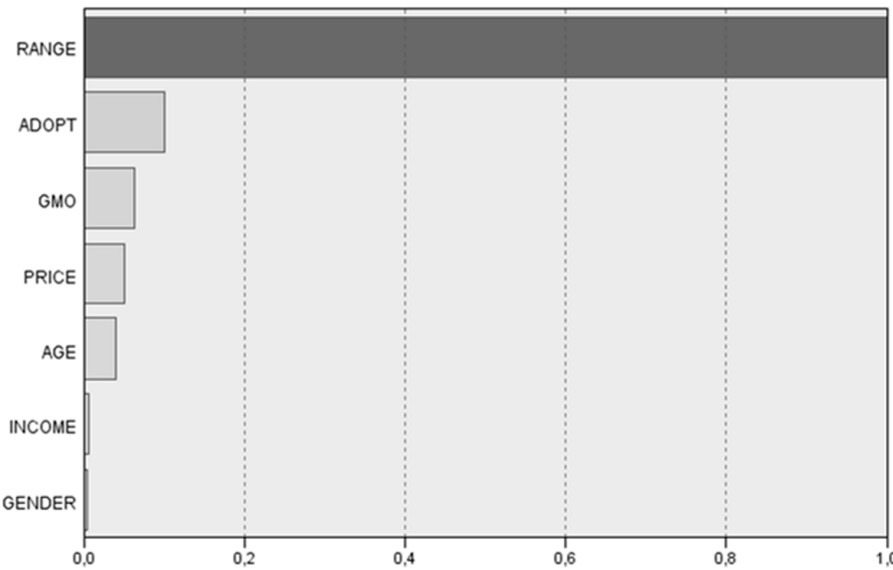

**Figure 2.** Predictor variable importance. Source: own conception using SPSS software.

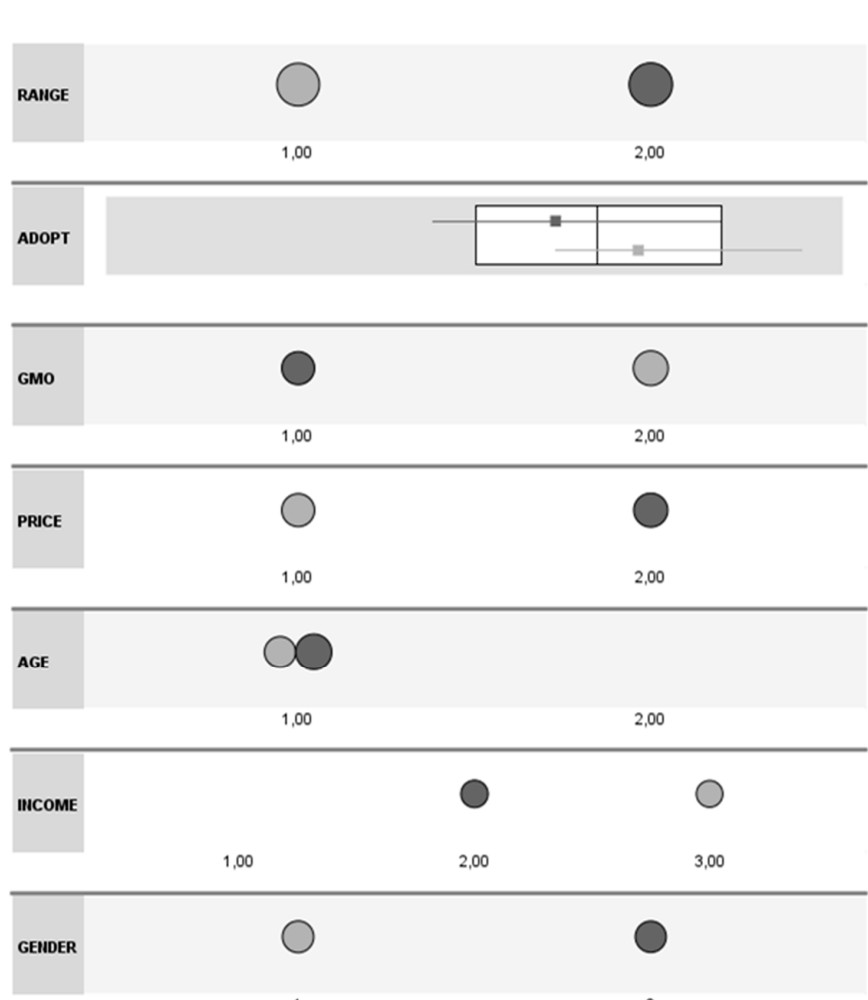

**Figure 3.** Cluster comparison. Note: The position of each circle indicates the predominant value (1 or 2) for the variable in that segment (1: light—Early adopters; 2: dark—Late adopters).

**Table 5.** Cluster comparison.

| Cluster 1 | | Cluster 2 | |
| --- | --- | --- | --- |
| Dominant Characteristics | Proportion (%) | Dominant Characteristics | Proportion (%) |
| Female | 53 | Male | 51.3 |
| Young | 52.4 | Young | 69.6 |
| High income | 37.6 | Medium income | 39.3 |
| High GMO | 65.1 | Low GMO | 57.6 |
| Low range | 97.9 | High range | 100 |
| Low price | 58.2 | High Price | 61.8 |
| Intention to adopt | 5.40/7 | Intention to adopt | 4.52/7 |

Source: own conception.

## 5. Discussion

This study makes several valuable contributions to existing theory and practice in marketing related specifically to the adoption of EVs. Current knowledge related to the adoption of EVs shows that it depends on a plethora of issues related to consumers, the vehicles themselves, transport infrastructure and government policy, though with very different levels of impact across countries [15,25,26]. As a result of this, the marketing of this technology requires a country-specific knowledge of the characteristics that define potential adopters of EVs in order to improve policymaking and strategic approaches aimed at developing a wider level of market share for EVs in view of reducing noxious gas emission levels related to road transport. Yet, few studies up to now have analysed the profile of potential early adopters of EVs in Spain, one of the largest countries in Europe in terms of population and one of the world's largest car manufacturers.

The results of this research have identified two consumer segments in terms of their intention to adopt EVs. The most important predictor variable in both segments was found to correspond to EVs' driving range. This finding is in line with earlier studies [19,38,80,97], though it emphasises the importance of this variable in the segmentation of this market as few studies up to now have identified this variable as the dominant characteristic governing early adopters' attitude towards EVs.

This study introduces a new variable in the development of a profile of adopters of EVs: green moral obligation. Earlier studies had shown that adopters of EVs tend to be concerned about the environment [17,32,50]. However, few studies had established a link between the adoption of EVs and customers' personal values and norms. Therefore, although the predictor capacity of GMO is not high, the results obtained in this study contribute to a better understanding of early adopters' motivations to purchase EVs.

The analysis of the data rendered by this study confirms the findings of earlier research, where price of EVs was shown to be a key barrier among their potential adopters [15,73,75,77]. Even if this variable is less important compared to the two outlined earlier (driving range and GMO), this study shows that it retains a significant impact on the market segmentation of adopters of EVs in Spain.

This study shows that socio-demographic characteristics, when considered at individual level, have a low influence on the segmentation of potential adopters of EVs. Although there do not appear to be major differences related to gender, it could be argued that consumers more likely to purchase EVs could be profiled in general terms as young women within the higher income band. This would appear to be in line with earlier studies [21,31,32], even if other research has shown that, generally, male customers tend to be more likely to purchase EVs [12,14,29,30]. In this particular study, the higher levels of concern about climate change among female customers in Spain may go some way to explain this [98]. Regarding the higher levels of income enjoyed by early adopters of EVs in Spain, the results

of this study are in line with those of earlier research [40–43], so this study confirms the importance of this variable in market segmentation strategies. Additionally, regarding consumers' age, there was a higher proportion of young people among respondents less likely to purchase EVs. Although this result contradicts those of earlier studies [16,18,32,99], this remains in line with the findings of some studies, which established a lower level of propensity to adopt EVs among young consumers with lower income or with higher mobility requirements than those of older consumers [37,38].

Finally, this study provides a contribution to existing knowledge on this topic by identifying the characteristics of two different groups of potential adopters of EVs. In the early adopters' group, the price and driving range of EVs are deemed as less important, whereas GMO is more important. In this group, female customers, young people and high-income customers prevail. On the other hand, the second group could be labelled as late adopters. This group is characterised by concerns among its customers with regards to the driving range of EVs and their price, with a lower level of GMO. In this late adopters' group, male and medium-income customers prevail, with a higher proportion of younger customers than in the early adopters' group. These findings provide a better understanding of the characteristics of potential adopters of EVs in the Spanish market, which had received little attention in earlier research studies compared to other countries.

This study also contributes to the marketing and commercialisation of EVs as it outlines a customer profile for early adopters of this product. The findings of this study will help to develop better communication and retail strategies aimed at potential adopters of EVs. For instance, in the case of early adopters it would be important to pay special attention to young women with a high level of income and place a specific emphasis on EV attributes related to environmental issues, including the lower $CO_2$ emissions of EVs or their low noise pollution levels. However, retail and communication strategies aimed at late adopters of EVs would have to target mainly male customers and use arguments that contribute to reducing the key barriers identified earlier. This could include, for instance, government incentives for the purchase of EVs or reducing the impact of price on purchase intentions by using marketing that emphasises the cost efficiency of EVs in terms of performance. Similarly, in order to reduce the negative perception of EVs' driving range, manufacturers of EVs could lobby governments for improvements in the network of charging points.

To conclude, this research study is subject to several limitations. Firstly, the research was performed using a sample of potential users of EVs. Further studies could contrast the findings of this study with a survey of existing owners of EVs. Secondly, this research has investigated the customer profile of early adopters of EVs using customer characteristics as well as vehicle attributes. Further studies could build on this by including other related variables, including charging point infrastructure and policy instruments for the promotion of EVs. Thirdly, variables such as age, GMO, driving range and price were re-coded into two categories to perform the cluster analysis. Although this technique allows for the creation of large groups, it can make it difficult to identify smaller consumer market segments. Therefore, it is suggested that further research should address this finer-grained level of analysis by creating different data analysis criteria to re-code these variables. Finally, the concept of EVs used in this study refers to battery-operated electric cars and plug-in hybrid electric cars. Further research in this field could explore separate profiles of early adopters either type of vehicle, which would help to elicit potential differences.

## 6. Conclusions

A good understanding of the profile of potential adopters of EVs is key for the development and implementation of initiatives aimed at growing current rates of adoption of these types of vehicles. This study has explored different factors to create a customer segmentation of Spanish consumers in terms of their readiness to adopt EVs. Based on a review of relevant literature on this topic, and in line with rational choice theory, this study has evaluated the predictive capacity of three socio-demographic characteristics (gender, age and income), one psychographic variable (green moral obligation), and two vehicle specific attributes (price and driving range). For this purpose, a two-stage cluster analysis has

been performed on data gathered through an online survey involving 404 respondents. The results of this analysis have shown that driving range was the most important factor and that, to a lesser extent, green moral obligation, price and age followed, in that order. This study concludes that customers most likely to purchase an EV in Spain are female customers, young people and high-income consumers. Similarly, this group had a higher level of green moral obligation and a less negative perception of issues related to EVs' driving range and price. On the other hand, customers with the lowest likelihood to purchase EVs are generally male with an average income, a low level of green moral obligation, and for whom driving range and the market price of EVs remain considerable obstacles to their adoption. As a result, this study contributes to a better understanding of the profile of early adopters of EVs in Spain and highlights the importance of considering green moral obligation, a hitherto neglected variable, as part of future research in this field.

**Author Contributions:** Conceptualization, methodology and data analysis, E.H.-C. and F.L.-C.; Writing—Original draft preparation, E.H.-C., S.M. and J.A.C.-S.; Writing—Review and editing, E.H.-C., S.M. and J.A.C.-S.; supervision, F.L.-C. All authors have read and agreed to the published version of the manuscript.

**Funding:** This research was funded by the Spanish Ministry of Science, Innovation and Universities, National R&D&I Plan and FEDER under grant B-SEJ-209-UGR18.

**Conflicts of Interest:** The authors declare no conflict of interest.

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
