# Peer review of "Potential Early Adopters of Hybrid and Electric Vehicles in Spain—Towards a Customer Profile"

_sustainability, doi:10.3390/su12114345_

Round 1

Reviewer 1 Report

The paper deals with an issue related to "Potential early adopters of hybrid and electric 2 vehicles in Spain – towards a customer profile", which can be considered in line with the scope of this journal. Despite the importance of the topic and the appropriateness of the adopted approach, the paper seems to have several main issues that must be solved, prior to being considered for publishing, such as:

- in point 2.1 "The role of individual-related characteristics in the adoption of EVs" of section 2 "Theoretical Background" it is assumed that Gender, Age and Income are the main characteristics corroborated by many works presented in Table 1. However, it would also be important, or even more interesting, to verify in the literature whether there are other characteristics different from those presented that could be applied in the specific context of Spain, instead of assuming a priori that these are the 3 most important characteristics;

- in point 2.2. The role of financial and technical attributes in the adoption of EVs, studies that integrate the issues of technological innovation and above all the functioning of the charging network should be considered, namely the fast charging points.

- in section 3. Research Method it is necessary to give more detailed information about the dissemination, as well as to explain ‘where’ and ‘how’ the questionnaires were done. Although some items are presented in Table 2 for evaluation, it is not clear whether the questionnaire was based only on these questions or whether it was a more comprehensive questionnaire. Thus, it would be important to make a presentation of the survey structure with the data groups raised in the investigation.

- in point 3.2 Sample, the information about the sample of this work is presented in some detail. But, more important than that is the level of significance of this sample, which is not presented. It is important to enhance that the authors are presenting a study for an entire country with only 404 questionnaires. Furthermore, what is the population considered? What is the sample stratification? What was the assessment of the sample's alignment with the population? These issues raise some doubts about the presented results, which correspond to the main weaknesses of the Research Method and this work. Besides, the definition of only 2 classes for age, GMO, Range and Price, is not very clear and needs clarification. Why didn’t the authors consider 3 or 4 classes for each of these items? And by the way, how was performed the calculation of just one indicator for each of the items, when they were evaluated by more than one question? Certainly, more classes in the variables could give different results than those presented, since they used the Clusters technique.

- In section 4. Results, it would be important to present more results from the statistical process, not just the "predictor variable importance". Besides, the paragraphs between lines 260 - 275 need to be corroborated by more statistical information, i.e., the results of the responses for each cluster (e.g. a Table), namely to better understand a cluster formed by 53% female respondents gives the idea that it is mostly set up by women, but in reality, it is almost 50/50 ... Finally, Figure 3 may be interesting from a graphical point of view, but it would be much more interesting and valuable to have values or percentages of individuals allocated by cluster to each variable.

Author Response

Please find detailed reponse from authors in file attached.

Reviewer 2 Report

The paper is appropriate and relevant  to the Sustainability Journal themes, providing a cluster analysis to profile early adopter of EVs in Spain.

Research objectives and methodologies are clear, and the paper is well structured.

In my opinion, you should specify better the segmentation of variables used in the study: you said that variables like GMO, Range and Price were re-coded into two categories in line with the average value obtained. Anyway, it may be useful to specify better those values and how they were divided into low and high.

Author Response

(The authors gave the same response as above.)

Reviewer 3 Report

After I read the paper, I have some remarks.

  1. My main concern regarding this paper is representativity of the sample for the whole Spain population. To be more specific, the author declares that the research was conducted at the Spain level (“The purpose of this study is to identify the customer profile of early adopters of EVs in Spain: one of Europe’s largest economies…”). Nevertheless, I don’t know if the sample is representative for the whole population of Spain. Just as an example, the author present the table 3, where ages of the sample is like this: 18-25 years (13.6%), 26-35 years old (27.5%), 36-45 years old (19.1%), 46-55 years old (18.3%), 56-65 years old (13.9%), older than 65 years (7.7%). But it is not clear if those percents are the same for the whole population of Spain (otherwise, the sample is not representative for it). And this concern is valid for every category within the table (“Employment status”, “Monthly income”, “Driving experience”, “Annual distance driven” etc.)
  2. Every figure in the paper must presents the source (if there is the result of the author, then it must be like this: “Source: author own conception / calculation, using software XXX”)
  3. The final part of the article must be rewritten taking into account that the Discussion must be merged with Results (therefore, the new section will be “Results & Discussion”); the “Conclusions” section must be alone, apart from other sections (in this section, the author may present the final thoughts and the limitation of the study).

Author Response

(The authors gave the same response as above.)

Reviewer 4 Report

Main concern is about the novelty of the proposed work. Authors should highlight major differences and findings from other similar studies. 

In spite of authors argue with "This study introduces a new variable in the development of a profile of adopters of EVs: green moral obligation". I see no clear pieces of evidence of this

Also results section (in my opinion the most important it is only 18 lines and 2 figures. Authors should give more information and details.

Methodology needs more details about data, how was processed towards major results and compare with other studies.

Major findings it is already known.

Author Response

(The authors gave the same response as above.)

Reviewer 5 Report

Interesting and original paper with depht introduction and reference; being a sort of survey, more attention should be paid to the choice and management of the sample and the context within which it takes place, the Authors themselves recognize the limit that greatly reduces its impact. A more in-depth profiling of the sample and its insertion, work and location in society should be addressed.

Author Response

(The authors gave the same response as above.)

Round 2

Reviewer 1 Report

The authors have some changes in responding to my previous objections and suggestions. They have added some parts of the paper in response to my criticism. Although I don’t agree with all their responses, I think they have handled my criticism adequately. In my understanding, the paper should be improved in terms of the results, and better explained how the data from the "market research company - Toluna" was obtained. Are they available on open bases? And in that case where they can be found.

Author Response

Please refer to file attached

Reviewer 4 Report

work needs improvments.

If authors argue that novelty is the work in spain but what is the difference to this study

Junquera, Beatriz, Blanca Moreno, and Roberto Álvarez. "Analyzing consumer attitudes towards electric vehicle purchasing intentions in Spain: Technological limitations and vehicle confidence." Technological Forecasting and Social Change 109 (2016): 6-14.

or this

Martínez-Lao, Juan, et al. "Electric vehicles in Spain: An overview of charging systems." Renewable and Sustainable Energy Reviews 77 (2017): 970-983.

both are not cited...

Also there are difference among study for EV and hibrids? Most of study is applied to EV? Authors should clarify on this point

Author Response

Please refer to file attached

Round 3

Reviewer 4 Report

Authors adressed my issues